# Persistence of metric biases in body representation during the body ownership illusion

**Min-Hee Seo[1], Jeh-Kwang Ryu[2], Byung-Cheol Kim[3], Sang-Bin Jeon[4], Kyoung-Min Lee[1]***

**1** Interdisciplinary Program in Cognitive Science, College of Humanities, Seoul National University, Seoul, South Korea, **2** Department of Physical Education, College of Education, Dongguk University, Seoul, South Korea, **3** Department of Game Software, School of Software Engineering, Joongbu University, Seoul, South Korea, **4** Department of Computer Science, College of Engineering, Yonsei University, Seoul, South Korea

* kminlee@snu.ac.kr

**Data Availability Statement:** The datasets generated and/or analyzed during the experiments are available in the Mendeley Data repository (https://data.mendeley.com/datasets/257zj3m2sr/1, DOI: 10.17632/257zj3m2sr.1).

## Abstract

Our perception of the body's metric is influenced by bias according to the axis, called the systematic metric bias in body representation. Systematic metric bias was first reported as Weber's illusion and observed in several parts of the body in various patterns. However, the systematic metric bias was not observed with a fake hand under the influence of the body ownership illusion during the line length judgment task. The lack of metric bias observed during the line length judgment task with a fake hand implies that the tactile modality occupies a relatively less dominant position than perception occurring through the real body. The change in weight between visual and tactile modalities during the body ownership illusion has not been adequately investigated yet, despite being a factor that influences the perception through body ownership illusion. Therefore, this study aimed to investigate whether the dominance of vision over tactile modality is prominent, regardless of the task type. To investigate whether visual dominance persists during the process of inducing body ownership illusion regardless of task type, we introduced spatial visuotactile incongruence (2 cm, 3 cm) in the longitudinal and transverse axes during the visuotactile localization tasks and measured the intensity of the body ownership illusion using a questionnaire. The results indicated that participants perceived smaller visuotactile incongruence when the discrepancy occurred in the transverse axis rather than in the longitudinal axis. The anisotropy in the tolerance of visuotactile incongruence implies the persistence of metric biases in body representation. The results suggest the need for further research regarding the factors influencing the weight of visual and tactile modalities.

## Introduction

Humans do not accurately perceive their body's size and height; our perception of the body's metric is influenced by bias according to the axis, called the systematic metric bias [1]. In the case of the hand, its width (transverse axis) is usually overestimated, while the length

**Funding:** This research was supported by a National Research Foundation (NRF) grant funded by the Korean government and received my KML (https://www.nrf.re.kr/index, NRF-2017M3C7A1047225). The funders had no role in study design, data collection and analysis, decision to publish, or preparation of the manuscript.

**Competing interests:** The authors have declared that no competing interests exist.

(longitudinal axis) is underestimated. The systematic metric bias of body representation was first reported as Weber's illusion in the late 1900s and observed in other parts of the body in various patterns [2–5].

In addition to the pattern difference, the degree of systematic metric bias changes according to the type of task. However, the bias is less prominent when the task is vision-oriented (template matching task), compared to when the task is touch-oriented (skin localization task) [6]. In the template matching task, the participants compared the width of their own hands visually with the picture of the hand, while the localization task requested them to indicate the perceived location of the tip of the index finger on the board occluding their hands to block vision. This difference in prominence is explained in terms of the different types of body representations involved in the tasks. Specifically, explicit body image is involved in vision-oriented tasks, whereas implicit body representation is involved in touch-oriented tasks. Although this explanation is based on the participation of a specific category of body representation, the authors consider it as a spectrum rather than a category. For example, the degree of systematic metric bias observed in the line length judgment task (LLJ task) is in the middle of the degree observed in skin localization and template matching tasks since the former task is more vision-oriented than the skin localization task, but less vision-oriented than the template matching task. During LLJ task, participants judged whether the length of displayed line was shorter or longer than the perceived length of their index finger. Therefore, the degree of systematic metric bias can be regarded as the results of weight assigned to each sensory modality, which are vision and touch in this case.

However, the systematic metric bias was not observed with a fake hand under the influence of body ownership illusion (BOI) during the LLJ task [7]. As BOI refers to the illusion that makes people misperceive objects as part of their own body, the disappearance of metric bias seems unnatural. The concept of body ownership refers to the perception that "my body belongs to me" [8, 9]. The BOI is induced when the visual stimulus to a target object spatially and temporally coincides with the tactile stimulus to one's body through simultaneous visuotactile stimuli [10]. It is also induced by matching proprioception instead of touch with vision [11–13]. Although previous studies reported that body ownership was successfully induced in objects that did not belong to one's body, it seems that inducing BOI does not guarantee the same perception as that of one's real body.

The fact that no metric bias was observed during the LLJ task with a fake hand implies an unexpected finding that visual modality occupies a relatively more dominant position compared to perception occurring through the real body. The question of interest here is whether the dominance of vision prevails regardless of task type in the case of BOI. If this is true, it implies that there is a limitation to the BOI induced by simultaneous visuotactile stimulation. It is possible that inducing BOI through simultaneous visuotactile stimulation cannot elicit interactions at the level of implicit body representation. Consequently, the task that was originally touch-oriented turned into a vision-oriented one. This can cause performance differences in action tasks because implicit body representation is mainly involved in body movement.

Contrarily, the disappearance of metric bias in fake hand can be a matter of difference in weight rather than a complete inability to interact at the level of implicit body representation. Another possibility is that the weight assigned to the tactile modality is temporarily decreased in the case of a fake hand with a BOI. Considering that the LLJ task is a more visually oriented task than the skin localization one, the metric bias may have disappeared due to the decrease in weight in the tactile modality. If this is true, the metric bias can reappear during the skin localization task. Therefore, the weight difference of the tactile modality has the possibility of

being replenished rather than undergoing irreversible change, and the fake hand restores the position of plausible substitution as a perceptual medium similar to a real body.

To investigate this hypothesis, we observed the systematic metric bias using a virtual hand model while performing the localization task. In the experiments, the localization task required participants to judge the spatial incongruence between visual and tactile stimuli of simultaneous visuotactile stimulation. We measured the perception of incongruence by the BOI intensity using a questionnaire adjusted to the virtual reality (VR) environment. During the experiments, we compared the strength of BOI among different conditions with 2 cm and 3 cm of longitudinal and transverse spatial incongruence between vision and touch during the synchronous visuotactile stimulation on the participant's palm. We expected the anisotropic diminution of BOI intensity if the systematic metric bias reappears during the localization task, as the length of the hand is usually underestimated because of the horizontally stretched form of implicit body representation due to somatotopic distortion [14]. To observe the bias by axis, we pressed a specific point on the palm instead of brushing, which mixes up the two axes.

## Experiment 1

### Materials and methods

**Participants.**　Seventeen right-handed participants were recruited from the community website of Seoul National University. One participant was excluded from the final analysis because of failure to induce BOI (mean age = 25.3 years; standard deviation = 3.5; seven women). All participants were notified of the relevant information, following which they provided written informed consent of their participation in this study. The experiment was approved by the Institutional Review Board of Seoul National University.

**Virtual reality system.**　Participants wore head-mounted displays (HMDs; Vive) to experience an immersive virtual reality (VR) environment. A hand-tracking device (Leap Motion) was attached to the front of the HMD. A haptic device (Geomagic Touch) was included in the system to induce BOI via synchronous visuotactile stimulation. The pen-shaped part of the haptic device appeared in VR as a cylindrical stick, and a tracking system built on the device traced its location. However, the pen's opaque body interrupted the hand tracking of the Leap Motion. Thus, as a solution, a 30-cm acrylic cylinder with a diameter of 5 mm was attached to the pen. The transparency of the cylinder was sufficient for avoiding tracking errors (Fig 1).

The virtual environment was created using the software Unity. The environment reproduced the typical ambiance of an office room. When participants turn on the HMD, they saw a scene where they were sitting in front of a desk. They could see a virtual hand moving synchronously with their own hand. The participants were instructed to place their hands on the desk with their palms facing up.

**Experimental design.**　The experiment was a 2×2 factorial, with a within-group design consisting of factors on the degree of spatial incongruence (2 cm and 3 cm) and direction of spatial incongruence (longitudinal and transverse). There were five conditions—four experimental conditions and one control condition—which provided tactile stimulation to the center of the palm without spatial discrepancy. Throughout the experiment, the participants were blinded to the condition. The order of condition presentation was randomized to minimize the influence of environmental factors.

The software was able to manipulate the position of the acrylic cylinder in VR by 1 cm in the transverse (the direction that crosses the palm from left to right) and longitudinal directions (the direction that crosses a palm from the finger to the wrist and is perpendicular to the transverse direction). To maintain the consistency of the axis, the participants were instructed not to move their hands once the experiment began. The participants used their non-

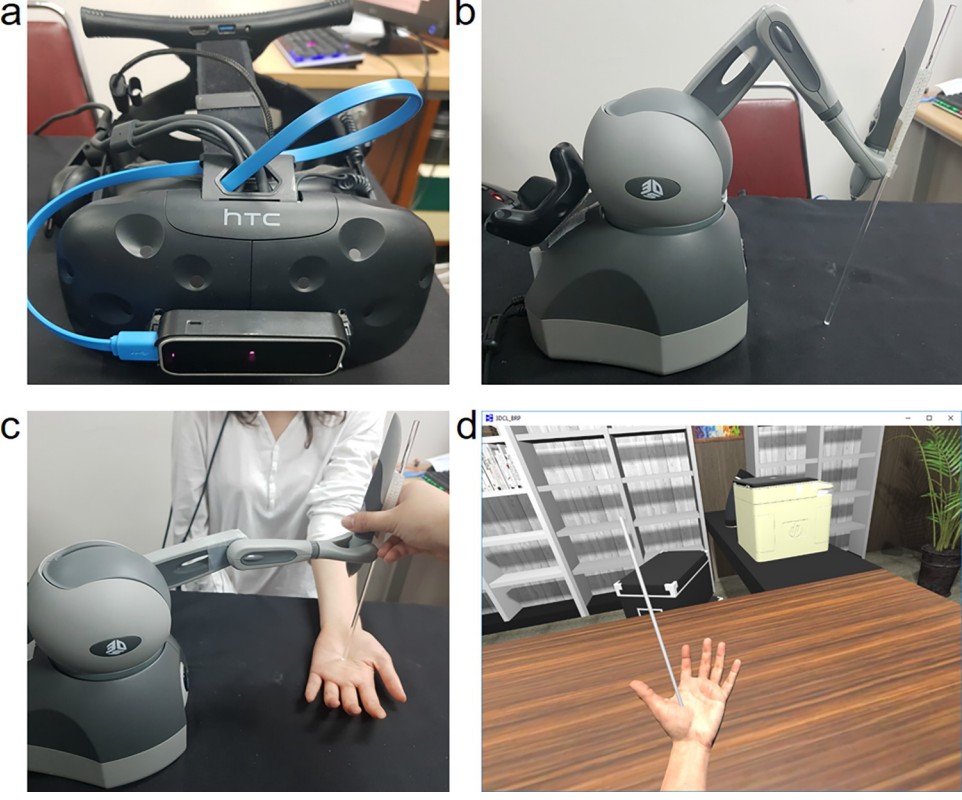

**Fig 1. Virtual reality (VR) system.** The above image shows the hardware of the VR system. Image *a* shows a head-mounted display with Leap Motion attachment. Image *b* shows a haptic device with an acrylic cylinder attached to the pen. The image below compares the actual (c) and VR scenes (d) of the synchronized VT stimulation.

dominant left hand for the experiment to consider the asymmetric ability of proprioceptive target matching [15, 16].

All experimental conditions fixed the degree of spatial incongruence to 2 cm or 3 cm, and the size of incongruence remained consistent throughout each condition by monitoring the existence of errors in the tracking device using dual monitors. The location of the tactile stimulation was maintained by marking the site of stimulation. The numerical value of spatial incongruence stems from previous studies on proprioceptive drift measured after the induction of BOI [17, 18]. Fig 2 shows the detailed stimulation site for each experimental condition.

**Questionnaire.** The questionnaire was designed to assess the intensity and quality of the BOI experience in VR based on the original questionnaire used for the rubber hand illusion by Botvinick and Cohen (1998) [10]. However, unlike the rubber hand illusion, the virtual hand can co-localize with the real hand. The questionnaire reflected the characteristics of the VR setup. The responses were scored on a 5-point Likert scale ranging from 1 (totally disagree) to 5 (totally agree). The questionnaire was thereafter translated into Korean. Two native Korean individuals, fluent in both, Korean and English, confirmed the validity of the translated questionnaire. Table 1 presents the English version of the questionnaire and the explanation of what they measure.

We measured not only the overall intensity of BOI (Q3) but also the perceived degree of spatial incongruence (Q1), referral of touch (Q2), a perceived difference in visual attributes (Q4), and the success in drift of hand location to compensate for spatial incongruence between vision and touch (Q5).

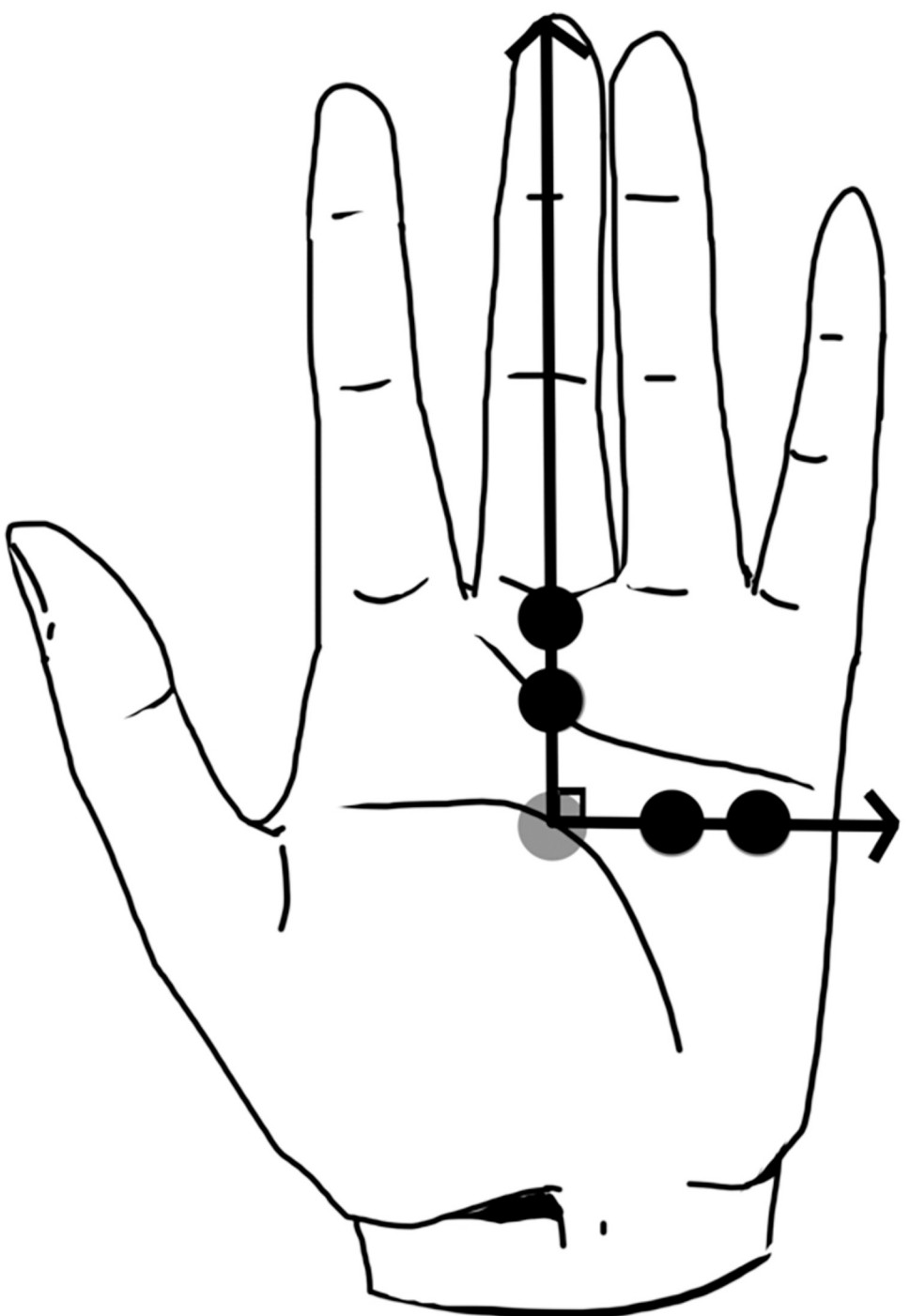

**Fig 2. Experimental conditions with different types of spatial discrepancy.** The vertical and horizontal axes represent the longitudinal and transverse directions, respectively. The dots depict the site of stimulation and degree of spatial incongruence (2 cm and 3 cm). The gray dot is the site of tactile stimulation, whereas the black dots refer to the site of virtual visual stimulation.

**Procedures.**    The VR system was calibrated prior to the experiment. The experimenter provided VT stimulation at the center of the participant's palm. The participants then answered the question of whether they felt a mismatch between the visual feedback and tactile

**Table 1. BOI questionnaire.**

| | | |
|---|---|---|
| Q1 | It seemed as if I were feeling the touch of the stick at the same location as where I saw the virtual hand touched | Perceived spatial congruence during simultaneous visuotactile stimulation |
| Q2 | It seemed as though the touch I felt was caused by the stick touching the virtual hand | Referral of the touch to the virtual hand |
| Q3 | I felt as if the virtual hands were my hands | The subjective intensity of overall BOI |
| Q4 | The virtual hand began to resemble my own hand, in terms of shape, skin tone, freckles, or other visual features | The feeling of visual similarity to the virtual hand |
| Q5 | It seemed as if my own hand was located on the site of the virtual hand | Feeling of proprioceptive congruence between the real and virtual hand. |

stimulus. The calibration process was continued until the participants reported no feeling of spatial incongruence.

At the beginning of the experiment, the participants closed their eyes while the experimenter manipulated the position of the acrylic cylinder to remain uninformed of the condition. They opened their eyes after the generation of proper spatial incongruence and 90 s of synchronous VT stimulation followed by spatial incongruence. When the stimulation ended, the participants reported their experience of BOI by verbally answering the questionnaire. This procedure was repeated for each condition. There was one trial for each condition, and it took about 45 minutes to complete the experiment.

## Results

Statistical analyses were performed using IBM SPSS Statistics version 25. The total sum of the questionnaire scores was calculated to quantify the overall intensity of experience related to the BOI. Because the data did not satisfy the normality assumption, the Friedman test was conducted to assess the difference in the experience of BOI depending on the degree and direction of spatial discrepancy. The Friedman test is a nonparametric alternative to the 2-way ANOVA. In this study, a statistically significant difference was found between the conditions ($\chi^2(4)$ = 25.960, $p < 0.001$, *Kandell′s W* = 0.406). Dunn's pairwise post-hoc test was performed with Bonferroni correction to avoid the problem of multiple comparisons. The median and interquartile ranges (IQR) of the data are shown in Fig 3 and Table 2.

It was found that the intensity level of BOI was not significantly different from that of the control condition when there were 2 cm (Z = 0.615, p>0.999) and 3 cm (Z = 1.957, p = 0.386) of longitudinal spatial discrepancies. In contrast, the intensity level of BOI significantly decreased from that of the control condition when there were 2 cm (Z = 3.466, p = 0.005, $\frac{|Z|}{\sqrt{N}}$ = 0.867) and 3 cm (Z = 4.025, p = 0.001, $\frac{|Z|}{\sqrt{N}}$ = 1.006) of transverse spatial discrepancies. The overall BOI score also significantly decreased from that of the condition with a 2-cm longitudinal spatial discrepancy when there were 2 cm (Z = 2.851, p = 0.044, $\frac{|Z|}{\sqrt{N}}$ = 0.713) and 3 cm (Z = 3.410, p = 0.006, $\frac{|Z|}{\sqrt{N}}$ = 0.853) of transverse spatial discrepancy. The tolerance limit for VT mismatch was anisotropic. The results are described in Table 3.

Wilcoxon signed-rank tests were conducted for each item of the questionnaire to assess the origin of the total score. Bonferroni correction was applied, with the significance level set at p<0.01. The mean score and standard deviation of each item in the questionnaire are showed in Table 4. The results revealed that the difference mainly stems from Questions 1, 3, and 5, which inquired about the location consistency and feeling of BOI toward a virtual hand. Table 5 describes the results in detail.

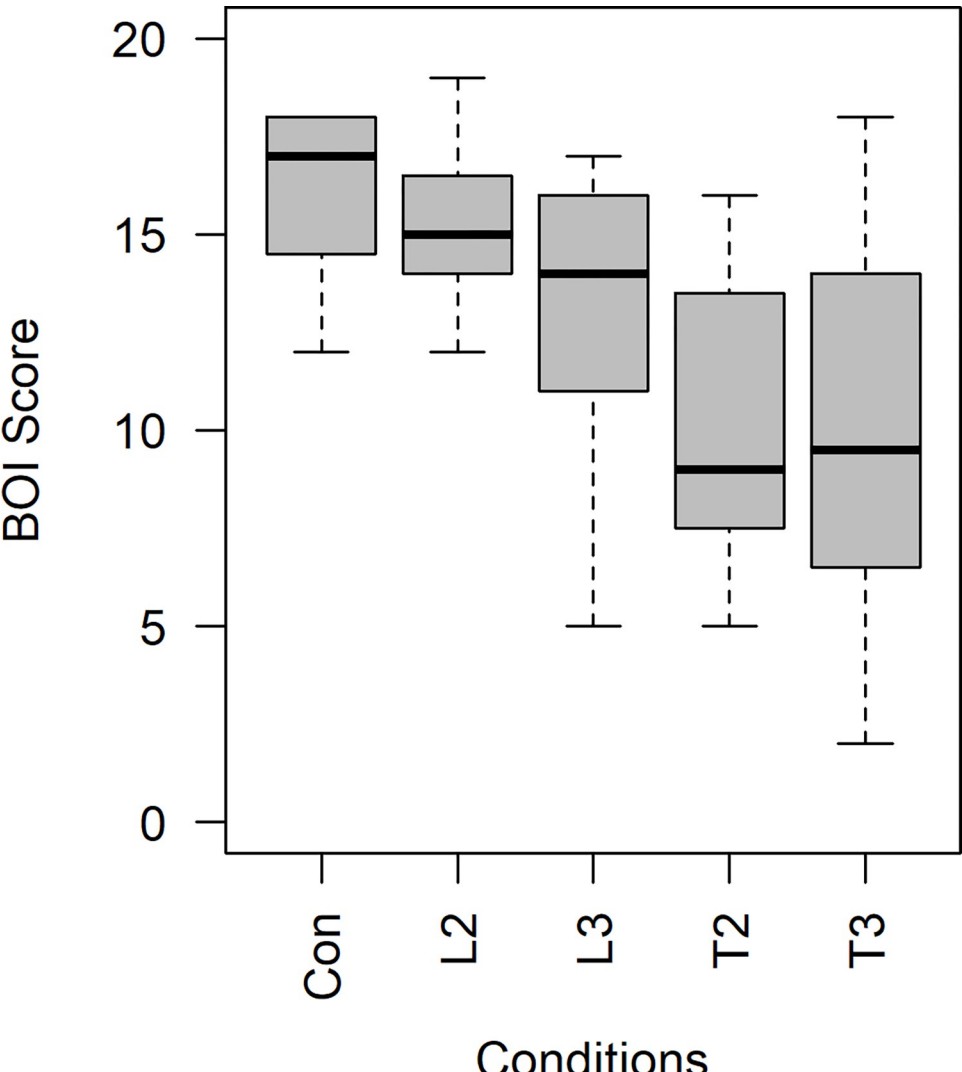

**Fig 3. Boxplot of BOI score for each condition.** Con: control condition; L: longitudinal; T: transverse; 2: 2 cm; 3: 3 cm.

## Discussion

In Experiment 1, the anisotropic diminution of the BOI questionnaire score was observed. However, it is possible that the posture of the participants' hands resulted in the depth

**Table 2. Median and IQR of the total questionnaire score.**

| Condition | N | Minimum | 25th percentile | Median | 75th percentile | Maximum |
|-----------|---|---------|-----------------|--------|-----------------|---------|
| Con | 16 | 12 | 14.25 | 17 | 18 | 18 |
| L2 | 16 | 12 | 14 | 15 | 16.75 | 19 |
| L3 | 16 | 5 | 11 | 14 | 16 | 17 |
| T2 | 16 | 5 | 7.25 | 9 | 13.75 | 16 |
| T3 | 16 | 2 | 6.25 | 9.5 | 14 | 18 |

*Note*. N: sample size, Abbreviation: Con: control condition; L: longitudinal; T: transverse; 2: 2 cm; 3: 3 cm

**Table 3. Friedman test of the total questionnaire score.**

| Condition | Test statistics | Standard error | Standard test statistics | Adjusted significance level | $\frac{|Z|}{\sqrt{N}}$ |
|---|---|---|---|---|---|
| Con-T2 | 1.938 | 0.559 | 3.466** | 0.005 | 0.867 |
| Con-L2 | 0.344 | 0.559 | 0.615 | >0.999 | 0.154 |
| Con-T3 | 2.250 | 0.559 | 4.025** | 0.001 | 1.006 |
| Con-L3 | 1.094 | 0.559 | 1.957 | 0.504 | 0.489 |
| L2-T2 | 1.594 | 0.559 | 2.851* | 0.044 | 0.713 |
| L2-T3 | 1.906 | 0.559 | 3.410** | 0.006 | 0.853 |
| L2-L3 | 0.750 | 0.559 | 1.342 | >0.999 | 0.336 |
| T2-T3 | 0.312 | 0.559 | 0.559 | >0.999 | 0.140 |
| T2-L3 | 0.844 | 0.559 | 1.509 | >0.999 | 0.377 |
| T3-L3 | 1.156 | 0.559 | 2.068 | 0.386 | 0.517 |

Abbreviation: Con: control condition; L: longitudinal; T: transverse; 2: 2 cm; 3: 3 cm

*p<.05

**p<.01

perception with insufficient pictorial depth cues in VR. Pictorial depth cues refer to monocular cues about depth information [19]. In VR, the underestimation of depth has been reported in previous studies due to the insufficient depth cues [20–22]. With the hand laid flat on the table, the participants had to perceive the depth in longitudinal condition, which is usually underestimated in a VR environment. This could have caused an underestimation of the distance in the longitudinal direction and showed an anisotropic decrease in the questionnaire score.

Additionally, the minute shaking of the virtual hand due to the tracking problem caused by the body of the experimenter and the acrylic cylinder blocking the base station may have resulted in inaccuracy in overall distance perception. We conducted a second experiment to eliminate the confounding factors. In the second experiment, the participants raised their hands from the surface of the desk and looked down at their hands vertically to minimize depth perception. Furthermore, the position of the virtual hand was fixed after the calibration of the hand posture and stick position to minimize tracking error.

## Experiment 2

### Materials and methods

**Participants.**   Thirty-four right-handed participants were recruited from the community website of Seoul National University. Three were excluded because of the failure of the

**Table 4. Descriptive statistics of each item in the questionnaire by condition.**

| Mean score and standard deviation of each item | | | | | |
|---|---|---|---|---|---|
| Item | Con | L2 | L3 | T2 | T3 |
| Q1 | 3.69±0.479 | 3.19±0.750 | 2.06±1.340 | 1.25±1.238 | 1.19±1.377 |
| Q2 | 3.50±0.730 | 3.25±0.577 | 3.00±1.095 | 2.50±1.095 | 2.38±1.408 |
| Q3 | 2.87±0.500 | 3.06±0.680 | 2.37±0.957 | 1.88±1.147 | 2.13±1.258 |
| Q4 | 2.50±0.730 | 2.63±0.806 | 2.63±0.885 | 2.13±0.806 | 1.69±0.946 |
| Q5 | 3.63±0.500 | 3.25±0.447 | 3.00±1.155 | 2.50±1.095 | 2.31±1.352 |

Abbreviation: Con: control condition; L: longitudinal; T: transverse; 2: 2 cm; 3: 3 cm

**Table 5. Wilcoxon signed-rank test of each questionnaire score.**

(a) Comparison between Con and T2

| Item | N | Z | P-value | $\frac{|Z|}{\sqrt{N}}$ |
|------|---|---|---------|--------|
| Q1 | 16 | -3.225* | 0.001 | 0.806 |
| Q2 | 16 | -2.347 | 0.019 | 0.587 |
| Q3 | 16 | -2.676* | 0.007 | 0.669 |
| Q4 | 16 | -1.234 | 0.217 | 0.309 |
| Q5 | 16 | -2.811* | 0.005 | 0.703 |

(b) Comparison between Con and T3

| Item | N | Z | P-value | $\frac{|Z|}{\sqrt{N}}$ |
|------|---|---|---------|--------|
| Q1 | 16 | -3.225* | 0.001 | 0.806 |
| Q2 | 16 | -2.162 | 0.031 | 0.541 |
| Q3 | 16 | -2.142 | 0.032 | 0.536 |
| Q4 | 16 | -2.506 | 0.012 | 0.627 |
| Q5 | 16 | -2.698* | 0.007 | 0.675 |

(c) Comparison between L2 and T2

| Item | N | Z | P-value | $\frac{|Z|}{\sqrt{N}}$ |
|------|---|---|---------|--------|
| Q1 | 16 | -3.169* | 0.002 | 0.792 |
| Q2 | 16 | -2.443 | 0.015 | 0.610 |
| Q3 | 16 | -2.992* | 0.003 | 0.748 |
| Q4 | 16 | -1.903 | 0.057 | 0.476 |
| Q5 | 16 | -2.489 | 0.013 | 0.622 |

(d) Comparison between L2 and T3

| Item | N | Z | P-value | $\frac{|Z|}{\sqrt{N}}$ |
|------|---|---|---------|--------|
| Q1 | 16 | -3.377* | 0.001 | 0.844 |
| Q2 | 16 | -2.226 | 0.026 | 0.557 |
| Q3 | 16 | -2.658* | 0.008 | 0.665 |
| Q4 | 16 | -2.373 | 0.018 | 0.593 |
| Q5 | 16 | -2.235 | 0.025 | 0.559 |

Abbreviation: Con: control condition; L: longitudinal; T: transverse; 2: 2 cm; 3: 3 cm

*p<.01

tracking system, and two because of the failure to induce BOI in the congruent condition (Q3 <2). Therefore, 29 participants were included in the final analysis (mean age = 25.7 years, SD = 3.6; 16 women). All participants were notified of the relevant information, following which they provided written informed consent. The experiment was approved by the Institutional Review Board of Seoul National University.

**Experimental design.** The experiment was a 2×2 factorial, with a within-group design consisting of factors on the degree of spatial incongruence (2 cm and 3 cm) and direction of spatial incongruence (longitudinal and transverse). There were five conditions—four experimental conditions and one control condition—which provided tactile stimulation at the center of the palm without spatial discrepancy. The VR system setup was the same as that used in Experiment 1. Throughout the experiment, the participants were blinded to the condition. The order of condition presentation was randomized to minimize the influence of environmental factors.

**Procedure.** The participants of Experiment 2 were instructed to place their arms close against a desk with their palms up to minimize the depth perception. The participants raised

their hands from the surface of the desk using their wrists and looked down at their own hands vertically, in a way that the horizontal line of sight was perpendicular to the longitudinal axis of the hands. They were instructed not to move after the commencement of the experiment. Moreover, instead of attaching an acrylic cylinder, we fixed the position of a virtual hand after the participants put their hands in the instructed posture. This was done to minimize errors in hand tracking. Finally, the location of 2 cm and 3 cm spatial discrepancies were marked on a virtual hand using colored circles. This scene was not visible to the participants. The mark was added to minimize the confounding factor of location change originating from minute shaking of the virtual world due to tracking error. The rest of the procedures were the same as in Experiment 1. There was one trial for each condition, and it took about 45 minutes to complete the experiment.

## Results

An anisotropic pattern in the tolerance limit of the VT mismatch was observed after changing the viewing angle. The Friedman test was conducted because the data did not satisfy the normality assumption. There was a statistically significant difference found between the conditions ($\chi^2$(4) = 57.550, p<0.001, *Kandall's W* = 0.558). Dunn's pairwise post-hoc test was performed with Bonferroni correction. The median and IQR are shown in Fig 4 and Table 6.

The BOI intensity level did not significantly vary from that of the control condition when the longitudinal spatial discrepancy was 2 cm (Z = 1.536, p>0.999). The difference was significant when the size of the discrepancy reached 3 cm (Z = 4.900, $p < 0.001$, $\frac{|Z|}{\sqrt{N}} = 0.910$). However, the BOI intensity level significantly decreased from that in the control condition when there were 2 cm (Z = 4.858, p $< 0.001, \frac{|Z|}{\sqrt{N}} = 0.902$) and 3 cm (Z = 6.145, p $< 0.001, \frac{|Z|}{\sqrt{N}} = 1.141$) of transverse spatial discrepancies. A significant difference was observed between the longitudinal and transverse conditions when there was a 2 cm spatial discrepancy (Z = -3.322, p = 0.009, $\frac{|Z|}{\sqrt{N}} = 0.617$). Table 7 shows the detailed results of Friedman test.

Wilcoxon signed-rank tests were conducted for each component of the questionnaire to assess the origin of the total score. Bonferroni correction was applied, with a significance level set at p<0.01. The mean score and standard deviation of each item in the questionnaire are described in Table 8. The results revealed that the difference mainly stems from Questions 1 and 2, which inquire about the location consistency and experience of VT integration. The scores of Questions 3 and 5 were significantly different when the spatial congruence was more prominent. The score for Question 4 was not significantly different in all conditions. The results of Wilcoxon signed–rank tests are described in Table 9.

## General discussion

In the first experiment, we used the virtual hand model to observe the reappearance of systematic metric bias during the localization task. In the second experiment, the posture of the hand was changed to eliminate the difference in pictorial cues in VR that could have influenced the depth perception and reproduced the results of the first experiment. According to the results, the feeling of body ownership in the virtual hand model was more attenuated when spatial incongruence between visual and tactile stimuli occurred in the transverse axis, rather than in the longitudinal axis. A spatial incongruence of 2 cm was sufficient to cause a significant decrease in BOI intensity when incongruence occurred along the transverse axis (Con-T2) but not along the longitudinal axis (Con-L2). In addition, there was a significant decrease in BOI intensity in the L2 condition as compared to the T2 condition, thus suggesting that the

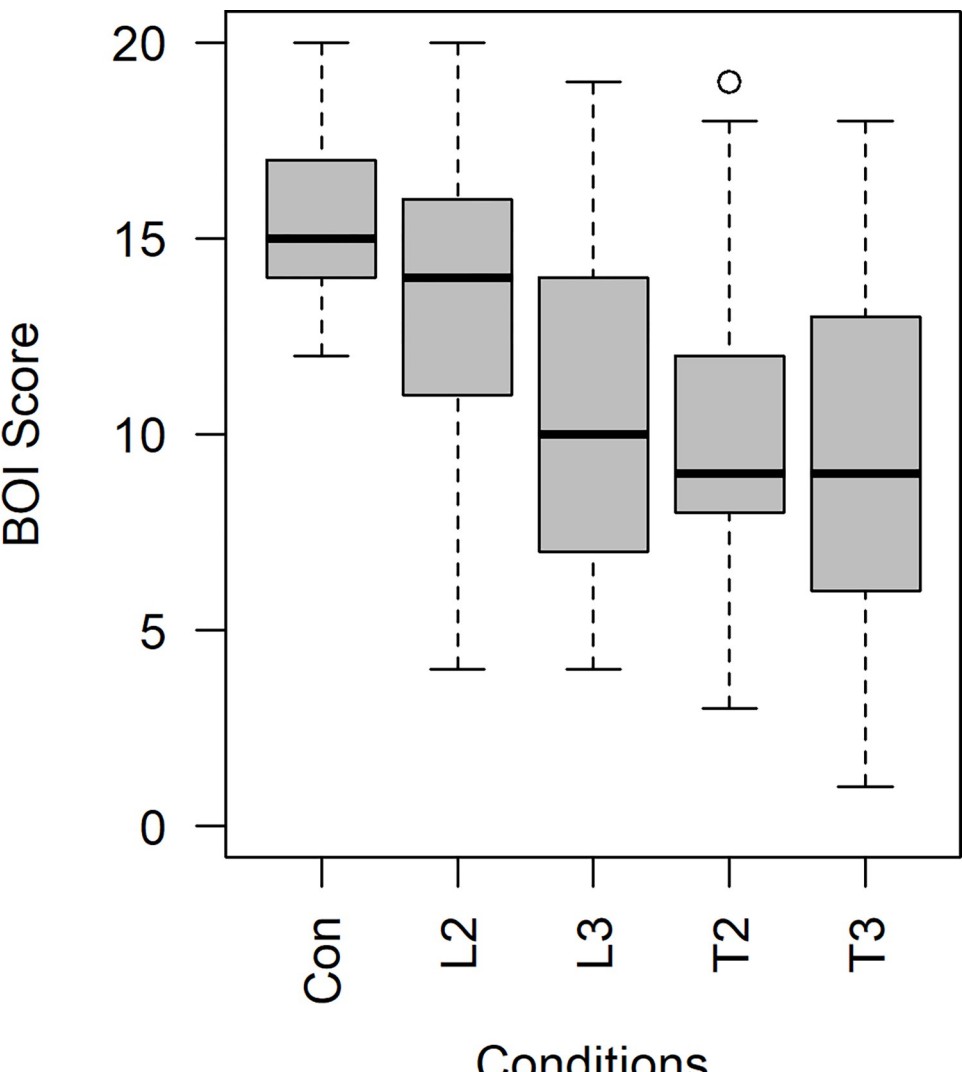

**Fig 4. Boxplot of the BOI score for each condition.** Abbreviation: Con: control condition; L: longitudinal; T: transverse; 2: 2cm; 3: 3cm.

perception of incongruence was more prominent in the transverse axis. No significant difference was found in BOI intensity when the spatial incongruence was over 3cm (T3-L3). This indicates that 3cm of spatial incongruence is sufficient to abort BOI in both, the transverse and

**Table 6. Median and IQR of the total questionnaire score.**

| Condition | N | Minimum | 25th percentile | Median | 75th percentile | Maximum |
|-----------|-----|---------|-----------------|--------|-----------------|---------|
| Con | 29 | 12 | 14 | 15 | 17 | 20 |
| L2 | 29 | 4 | 10.5 | 14 | 16 | 20 |
| L3 | 29 | 4 | 6 | 10 | 14.5 | 19 |
| T2 | 29 | 3 | 8 | 9 | 12 | 19 |
| T3 | 29 | 1 | 6 | 9 | 13 | 18 |

Abbreviation: Con: control condition; L: longitudinal; T: transverse; 2: 2cm; 3: 3cm.

**Table 7. Friedman test of the total score.**

| Condition | Test statistics | Standard error | Standard test statistics | Adjusted significance level | $\frac{|Z|}{\sqrt{N}}$ |
|---|---|---|---|---|---|
| Con-T2 | 2.017 | 0.415 | 4.858** | <0.001 | 0.902 |
| Con-L2 | 0.638 | 0.415 | 1.536 | >0.999 | 0.285 |
| Con-T3 | 2.552 | 0.415 | 6.145** | <0.001 | 1.141 |
| Con-L3 | 2.034 | 0.415 | 4.900** | <0.001 | 0.910 |
| L2-T2 | -1.379 | 0.415 | -3.322** | 0.009 | 0.617 |
| L2-T3 | -1.914 | 0.415 | -4.609** | <0.001 | 0.856 |
| L2-L3 | 1.397 | 0.415 | 3.363** | 0.008 | 0.624 |
| T2-T3 | 0.534 | 0.415 | 1.287 | >0.999 | 0.239 |
| T2-L3 | 0.017 | 0.415 | 0.042 | >0.999 | 0.008 |
| T3-L3 | -0.517 | 0.415 | -1.246 | >0.999 | 0.231 |

Abbreviation: Con: control condition; L: longitudinal; T: transverse; 2: 2cm; 3: 3cm

*p<.05

**p<.01

longitudinal axis. This anisotropic diminution is similar to the predicted pattern of incongruence perception under systematic metric bias because of the participation of implicit body representations. We predicted that horizontally stretched body representation due to somatotopic distortions may lead to a decreased tolerance of spatial visuotactile incongruence in the transverse direction.

In this study, we measured the overall intensity of BOI (Q3) as well as the perceived degree of spatial incongruence (Q1), referral of touch (Q2), a perceived difference in visual attributes (Q4), and the success of drift of the hand location to compensate for spatial incongruence between vision and touch, if it exists (Q5). By considering multiple factors of incongruence perception, the total score of the questionnaire is robust to the inconsistent report originating from the intrinsic ambiguity in quantifying the feeling of body ownership.

We also clarified that the anisotropy did not originate from differences in the pictorial cues such as interposition, which is involved in visual depth perception. Although there are reports about the underestimation of evaluation of distance in VR [23], a difference in the underestimation rate by axis was not observed. Moreover, no significant underestimation of distance for under 300 cm was observed [24]. Therefore, considering the previous studies' results, the case of anisotropic underestimation of visual distance is unlikely since our study only included the evaluation of short distances—2 cm and 3 cm.

However, the differences found in the results between the first and the second experiment need to be addressed. First, the BOI score dropped in the second experiment. The BOI score of

**Table 8. Descriptive statistics of each item in the questionnaire by condition.**

| Mean score and standard deviation of each item | | | | | |
|---|---|---|---|---|---|
| Item | Con | L2 | L3 | T2 | T3 |
| Q1 | 3.52±0.574 | 2.62±1.115 | 1.38±1.208 | 1.07±1.100 | 0.76±0.988 |
| Q2 | 3.03±0.680 | 3.00±0.926 | 2.07±1.100 | 2.10±1.235 | 1.76±1.185 |
| Q3 | 3.03±0.626 | 2.55±1.152 | 2.07±1.193 | 2.24±1.057 | 2.07±1.163 |
| Q4 | 2.38±1.115 | 2.24±1.327 | 2.17±1.284 | 2.03±1.239 | 2.00±1.225 |
| Q5 | 3.48±0.574 | 2.97±1.180 | 2.79±1.082 | 2.66±1.143 | 2.79±1.013 |

Abbreviation: Con: control condition; L: longitudinal; T: transverse; 2: 2 cm; 3: 3 cm

**Table 9. Wilcoxon signed-rank test of each questionnaire score.**

(a) Comparison between Con and T2

| Item | N | Z | P-value | $\frac{|Z|}{\sqrt{N}}$ |
|---|---|---|---|---|
| Q1 | 29 | -4.670* | <0.001 | 0.867 |
| Q2 | 29 | -3.559* | 0.003 | 0.661 |
| Q3 | 29 | -3.035* | 0.002 | 0.564 |
| Q4 | 29 | -1.995 | 0.046 | 0.370 |
| Q5 | 29 | -3.529* | <0.001 | 0.655 |

(b) Comparison between Con and T3

| Item | N | Z | P-value | $\frac{|Z|}{\sqrt{N}}$ |
|---|---|---|---|---|
| Q1 | 29 | -4.679* | <0.001 | 0.869 |
| Q2 | 29 | -4.011* | <0.001 | 0.745 |
| Q3 | 29 | -3.370* | 0.001 | 0.626 |
| Q4 | 29 | -1.713 | 0.087 | 0.318 |
| Q5 | 29 | -3.207* | 0.001 | 0.596 |

(c) Comparison between Con and L2

| Item | N | Z | P-value | $\frac{|Z|}{\sqrt{N}}$ |
|---|---|---|---|---|
| Q1 | 29 | -3.714* | <0.001 | 0.690 |
| Q2 | 29 | -0.354 | 0.724 | 0.066 |
| Q3 | 29 | -2.288 | 0.022 | 0.425 |
| Q4 | 29 | -0.821 | 0.412 | 0.152 |
| Q5 | 29 | -2.289 | 0.022 | 0.425 |

(d) Comparison between Con and L3

| Item | N | Z | P-value | $\frac{|Z|}{\sqrt{N}}$ |
|---|---|---|---|---|
| Q1 | 29 | -4.419* | <0.001 | 0.821 |
| Q2 | 29 | -3.910* | <0.001 | 0.726 |
| Q3 | 29 | -3.381* | 0.001 | 0.628 |
| Q4 | 29 | -1.261 | 0.207 | 0.234 |
| Q5 | 29 | -3.337* | 0.001 | 0.620 |

(e) Comparison between T3 and L2

| Item | N | Z | P-value | $\frac{|Z|}{\sqrt{N}}$ |
|---|---|---|---|---|
| Q1 | 29 | 4.417* | <0.001 | 0.820 |
| Q2 | 29 | 3.878* | <0.001 | 0.720 |
| Q3 | 29 | -1.701 | 0.089 | 0.316 |
| Q4 | 29 | -1.393 | 0.163 | 0.259 |
| Q5 | 29 | -1.099 | 0.272 | 0.204 |

(f) Comparison between L2 and L3

| Item | N | Z | P-value | $\frac{|Z|}{\sqrt{N}}$ |
|---|---|---|---|---|
| Q1 | 29 | -3.690* | <0.001 | 0.685 |
| Q2 | 29 | -3.352* | 0.001 | 0.622 |
| Q3 | 29 | -2.098 | 0.036 | 0.390 |
| Q4 | 29 | -0.615 | 0.539 | 0.114 |
| Q5 | 29 | -0.996 | 0.319 | 0.185 |

(g) Comparison between L2 and T2

| Item | N | Z | P-value | $\frac{|Z|}{\sqrt{N}}$ |
|---|---|---|---|---|
| Q1 | 29 | -4.107* | <0.001 | 0.763 |
| Q2 | 29 | -3.430* | 0.001 | 0.637 |
| Q3 | 29 | -1.613 | 0.107 | 0.300 |

(*Continued*)

**Table 9.** (Continued)

| Q4 | 29 | -1.604 | 0.109 | 0.298 |
| Q5 | 29 | -1.671 | 0.095 | 0.310 |

Abbreviation: Con: control condition; L: longitudinal; T: transverse; 2: 2 cm; 3: 3 cm

*p<0.01

L3 did not decrease significantly from the control condition in the first experiment; however, there was a significant decrease in the second experiment. In addition, the size of the difference between T3 and L3 decreased in the second experiment. These differences indicate that the underestimation of depth in VR truly existed in experiment 1. When the participants laid their hands flat on the table, they perceived the depth as shorter when compared to the when they vertically looked down at their hands. This underestimation of depth might have been caused by the insufficient pictorial depth cues in VR.

The second difference was that the individual difference appeared more prominently at the L2 condition in the second experiment. Consequently, the degree of anisotropy decreased in the second experiment. However, we interpreted that the score variance at the L2 condition was too small in the first experiment due to the inaccurate depth perception. In the first experiment, there was no difference between the L2 condition and the control condition, since the distance of the location on the hand touched by the virtual stick was underestimated like in the case of the L3 condition. Consequently, the L2 condition showed an insignificant difference from the control condition which introduced no distance difference. Once the difference was perceived between the sites of touching, the individual difference in the 2-point discrimination threshold [25] could then be the origin of variance in BOI score under experimental conditions. A higher 2-point discrimination threshold could have led to the insensitivity of difference in perception between the sites of touching.

Despite the differences, the anisotropy in the T2-L2 comparison was maintained after the elimination of confounding factors that induced the underestimation of distance in the longitudinal direction, which supports the conclusion that the anisotropy was not caused by inaccurate depth perception.

Upon analyzing each item in the questionnaire of experiment 2, we found that the difference in the total score in the two 2 cm conditions (T2, L2) mainly originated from Q1 and Q2, both of which reflect the perceived spatial incongruence and referral of touch. The overall intensity of body ownership, which was assessed by Q3, was not significantly different between T2 and L2. However, there was a tendency for a decreased Q3 score in T2 compared to L2. In addition, the overall intensity of body ownership was significantly different from the congruent condition when the comparison was made with T2; however, the significant difference disappeared when the comparison was made with L2. In Q2, participants with 2 cm of longitudinal incongruence reported that although they felt a spatial mismatch between vision and touch, they felt that the stimulus was caused by the stick in VR. In contrast, participants with 2 cm of transverse incongruence reported a more prominent degree of spatial mismatch and felt that the stimulus was not caused by the stick in VR, which suggest that there was a failure to bind the stimulus and its visual origin because of more severely perceived spatial incongruence. From this perspective, it is plausible to say that there is an anisotropic tolerance of spatial incongruence by axis owing to the systematic metric bias. Therefore, the dominance of vision across task types during the generation of BOI was not observed in our study.

Regarding the methodological aspects, there can be a concern of subjectivity as the intensity of BOI was measured only using the questionnaire. However, it is not plausible that the specific

pattern of anisotropy in the tolerance of intermodal incongruence and the subsequent diminution of BOI resulted from subjectivity caused by self-reporting. Moreover, several studies have reported that proprioceptive drift, which is considered a classic measurement of BOI intensity, does not reflect the overall BOI intensity [17]. Proprioceptive drift generally relies only on visuoproprioceptive integration, which was controlled in our experimental design. Furthermore, the possible occurrence of proprioceptive drift by spatially incongruent visuotactile stimuli was reflected in the questionnaire by Q5.

To summarize, it is plausible to insist that we observed a systematic metric bias with the virtual hand model during the localization task. The results indicate that the disappearance of systematic metric bias during the LLJ task was caused by the relative weight decrease in the tactile modality to vision, and not the inability to interact at the level of implicit body representation. The decreased weight of the tactile modality was insufficient for inducing metric bias in the LLJ task, but was sufficient for inducing it in the localization task.

The reason for the decreased weight on the tactile modality can be explained in terms of participants' expectations. During the experiment, the participants did not expect the touching sensation to result from the virtual hand model. This top-down prior knowledge may have influenced weight calculation between the visual and tactile modalities. The statistically optimal integration theory supports the influence of prior knowledge as an index of reliability measurement for each sensory modality [26–28].

According to the theory, the weight of the tactile modality can be recovered if the reliability is restored to the extent of the real body. Several factors other than visual similarity and noise level of tactile stimuli can increase the reliability of tactile modality during BOI. Reliability may increase when participants use the fake hand without serious problems for a longer duration. On the contrary, it is possible that the reliability of tactile modality cannot be recovered to the level of the real hand, because of prior knowledge of the VR system. The influence of top-down and bottom-up factors on the weight calculation of sensory modality during BOI is an interesting topic for future research.

Another possibility to explore is that the anisotropic pattern in the perception of spatial visuotactile incongruence can work as an index for measuring the intensity of BOI. The self-report questionnaire has a limitation in that it can only measure the consciously reportable aspects of BOI [29]. Although the proprioceptive drift is considered a classical supplementation, the limitation is that it considers only the aspects of proprioception. If the degree of systematic metric bias observed at the BOI target correlates with the belief of the target being the origin of sensory stimulation, it can be considered a quantitative index for measuring the top-down aspects of BOI intensity.

## Acknowledgments

The technical assistance provided by student researcher Sun-Kyue Kim is appreciated.

## Author Contributions

**Conceptualization:** Jeh-Kwang Ryu, Byung-Cheol Kim, Kyoung-Min Lee.

**Data curation:** Min-Hee Seo.

**Formal analysis:** Min-Hee Seo.

**Funding acquisition:** Kyoung-Min Lee.

**Investigation:** Min-Hee Seo.

**Methodology:** Min-Hee Seo, Jeh-Kwang Ryu, Byung-Cheol Kim, Sang-Bin Jeon, Kyoung-Min Lee.

**Project administration:** Min-Hee Seo, Jeh-Kwang Ryu, Byung-Cheol Kim, Kyoung-Min Lee.

**Software:** Byung-Cheol Kim, Sang-Bin Jeon.

**Supervision:** Min-Hee Seo, Jeh-Kwang Ryu, Kyoung-Min Lee.

**Validation:** Min-Hee Seo.

**Visualization:** Min-Hee Seo.

**Writing – original draft:** Min-Hee Seo.

**Writing – review & editing:** Min-Hee Seo, Byung-Cheol Kim, Kyoung-Min Lee.

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
