## [Decision Letter · Decision Letter 0]

18 Mar 2022

PONE-D-22-04614Persistence of Metric Biases in Body Representation during the Body Ownership IllusionPLOS ONE

Dear Dr. Lee,

Thank you for submitting your manuscript to PLOS ONE. After careful consideration, we feel that it has merit but does not fully meet PLOS ONE’s publication criteria as it currently stands. Therefore, we invite you to submit a revised version of the manuscript that addresses the points raised during the review process.

Two expert reviewers have assessed your manuscript. Both reviewers found your study to be interesting and worthwhile. Reviewer 1 provides some helpful suggestions that will improve the clarity of your manuscript. Reviewer 2 has some more critical concerns regarding the sample sizes and the questionnaire methods you employed. Both reviewers note that the differences between your Experiments 1 and 2 need to be explained and discussed in more detail. Overall, I believe it should be possible for you to address all reviewer comments, and I look forward to receiving your revised manuscript.

We look forward to receiving your revised manuscript.

Kind regards,

Guido Maiello

Academic Editor

PLOS ONE

Journal Requirements:

Reviewers' comments:

Reviewer's Responses to Questions

**Comments to the Author**

1. Is the manuscript technically sound, and do the data support the conclusions?

Reviewer #1: Yes

Reviewer #2: Partly

2. Has the statistical analysis been performed appropriately and rigorously? 

Reviewer #1: Yes

Reviewer #2: Yes

3. Have the authors made all data underlying the findings in their manuscript fully available?

Reviewer #1: Yes

Reviewer #2: No

4. Is the manuscript presented in an intelligible fashion and written in standard English?

Reviewer #1: Yes

Reviewer #2: Yes

5. Review Comments to the Author

Reviewer #1: The paper by Seo and colleagues proposes the research findings testing whether the dominance of vision over tactile modality is prominent during the process of inducing body ownership illusion, regardless of task type. Using a VR environment, the authors introduced spatial visuotactile incongruence (2 cm, 3 cm) in the longitudinal and transverse axes during a visuotactile localization task. Results indicated that the feeling of body ownership in the virtual hand model was more attenuated when spatial incongruence between visual and tactile stimuli occurred in the transverse axis, rather than in the longitudinal one. The authors suggest that the anisotropy in the tolerance of visuotactile incongruence may imply the persistence of metric biases in body representation. I think the study is interesting, novel and well-conducted. The analyses are appropriate and sound. I only have a few minor suggestions that I hope may improve the quality of the manuscript.

- Line 51: The authors observe that the systematic metric bias might change as a function of the task type, such as template matching task, skin localization task or line length judgment task. Although they also cite some relevant references for these tasks, I believe that adding a few lines briefly describing them may help readers that are not familiar with these kinds of paradigms.

- Line 223: I think that the rationale behind Experiment 2 may be better clarified. It seems that the reason for conducting a second experiment was to eliminate some confounding factors related to depth perception in a VR system. However, I think this point may be expanded particularly for readers who are not experts in VR.

- Related to the previous point: (line 243) I find it a bit difficult to understand in which way the procedure of Experiment 2 differs from that of the first experiment. Could the author better clarify this aspect?

- Line 235: Did participants of Experiment 2 take part also in Experiment 1? Alternatively, are the two experiments conducted on different samples? Please specify.

- Results: Please report not only the significance but also the effect size for all the analyses.

- Figure 3: There is a mistake. It should be T2 and T3 and not C2 and C3.

Reviewer #2: Seo and colleagues investigate whether the perception of body ownership of a fake hand in VR is subject to a metric bias in body representation. Classic studies have reported an anisotropic tolerance for visuotactile mismatches, such that participants are more sensitive to mismatches along the longitudinal axis compared to the transverse. More recently, however, this anisotropy has been questioned. Seo et al. propose that task differences may be the reason for these different results and conjecture that the bias should be present in tasks where the tactile modality is relied on heavily. They test this conjecture in a VR experiment designed to (a) give participants a feeling of ownership of a virtual hand and (b) test whether this persists under visuotactile mismatches.

They indeed find that incongruences between seen and felt locations of a touch to the palm result in more strongly reduced perception of body ownership when along the longitudinal compared to the transverse axis, in line with known biases in tactile perception.

The authors make convincing case for why their experiments could advance our understanding of the body-ownership illusion and of body representation in different modalities in general. However, I think their data are much less clear than the manuscript currently suggests - this need not be a problem per se, but it does need to be discussed clearly and described in more detail.

Major comments:

1. Frankly, rather unequal sample sizes along with a just-significant difference in a key comparison in the experiment with the larger sample (exp.2, L2-T2) do not make me confident that these results are robust. I would feel much more at ease if there was some rationale for these sample sizes, as well as a more even discussion of the findings in experiment 2, i.e., not just of the significant difference L2-T2, but also of what the non-significant difference L3-T3 might mean.

2. Perhaps I missed it, but I would ask the authors to add a discussion of the difference between the two experiments. Notably, the much less clear anisotropy in the second experiment - meant to be a methodological improvement over the first - should be explained or at least touched upon. I think such a discussion on a more general level is missing and would add more value than the present discussion of many differences between single items and conditions (ll. 311-325) that may or may not mean much and where to me, it was not always clear which experiment was referred to. If the data do not paint a clear picture that is not itself a problem, but it needs to be made clear.

Minor/specific comments:

3. The fact that bias and perception of incongruence were assessed with a questionnaire took me a bit by surprise after reading the abstract and introduction. Given that different tasks are discussed at length early on, it might help clarifying what each of those measures specifically.

4. How long was each experiment? Was there one trial per condition, or multiple?

5. Providing mean scores per item, with variability, would be much preferred over the pairwise comparisons for each question that are currently given in tables 4 and 7.

Similarly, these scores should be provided in the data file, which currently contains only aggregate values.

6. P-values of precisely 0 or 1 should be reported as > .001 and < .999, respectively.

6. PLOS authors have the option to publish the peer review history of their article (what does this mean?). If published, this will include your full peer review and any attached files.

Reviewer #1: **Yes: **Andrea Ciricugno

Reviewer #2: **Yes: **Karl Kopiske

---

## [Author Response · Author response to Decision Letter 0]

8 Jun 2022

We thank you and the reviewers for your thoughtful suggestions and insights. The manuscript has been rechecked and the necessary changes have been made in accordance with the reviewers’ suggestions. The responses to all comments have been prepared as a word file. The responses attached below are the contents of the file submitted. 

Reviewer #1 

*Thank you for your valuable comments. Parts of the results of the manuscript were mixed with old version of manuscript. We apologize for the inattentiveness. We have revised the results section to ensure that there are no errors in the numerical values. No significant changes were made to the results and discussion sections. 

- Line 51: The authors observe that the systematic metric bias might change as a function of the task type, such as template matching task, skin localization task or line length judgment task. Although they also cite some relevant references for these tasks, I believe that adding a few lines briefly describing them may help readers that are not familiar with these kinds of paradigms.

- Response: Thank you for your comment. We added an explanation describing the procedure of the template matching task, skin localization task (line 54-56), and line length judgment task (line 64-66) in the manuscript. The line numbers are based on clear version of the manuscript.

“In the template matching task, the participants compared the width of their own hands visually with the picture of the hand, while the localization task requested them to indicate the perceived location of the tip of the index finger on the board occluding their hands to block vision.” 

“During the LLJ task, the participants judged whether the length of the displayed line was shorter or longer than the perceived length of their index finger.”

- Line 223: I think that the rationale behind Experiment 2 may be better clarified. It seems that the reason for conducting a second experiment was to eliminate some confounding factors related to depth perception in a VR system. However, I think this point may be expanded particularly for readers who are not experts in VR.

- Response: Thank you for your comment. We have added the reason for conducting a second experiment in the discussion section of Experiment 1. The second experiment was conducted to eliminate two confounding factors related to VR. 

First, the pattern of anisotropy can be confounded by the factors influencing the depth perception in VR. Specifically, when the hand of participants was laid in a flat position, the perception of distance in a longitudinal direction can be underestimated because of the differences in VR as compared to a real scene. In experiment 1, participants laid their hands flat on the table. However, this posture can cause an underestimation of distance perception in the longitudinal axis due to the shortcomings of the VR environment. For the accurate distance perception in the longitudinal axis, the depth cues should be presented like in a real-world scene. Due to the technological issue, participants in VR did not experience the full gamut of depth cues that were available in viewing a real world. The underestimation of depth in VR has been reported in previous studies (20-22).

Second, the minute shaking of the virtual hand due to the tracking problem that was caused by the body of the experimenter and the acrylic cylinder blocking the base station may have influenced the accuracy of distance perception. 

To eliminate the confounding factors, we conducted a second experiment. In the second experiment, participants raised their hands from the table and looked down at their own hands vertically. The horizontal line of sight was perpendicular to the longitudinal axis of the hands. In addition, the position of the virtual hand was fixed after the start position of the stick was ready to minimize the tracking error.

- Related to the previous point: (line 243) I find it a bit difficult to understand in which way the procedure of Experiment 2 differs from that of the first experiment. Could the author better clarify this aspect?

- Response: Thank you for your valuable comment. The difference in procedure in Experiment 2 was clarified by adding the reason for the posture change to the participants’ hands in the discussion section of Experiment 1. Unlike Experiment 1, participants raised their hands from the surface of the desk and looked down at their own hands to minimize the influence of the depth perception. The procedure of Experiment 2 was described in more detail by adding the procedure section.

- Line 235: Did participants of Experiment 2 take part also in Experiment 1? Alternatively, are the two experiments conducted on different samples? Please specify.

- Response: Thank you for your comment. New participants were recruited for Experiment 2, therefore, they were not a part of Experiment 1 and the sample for both experiments were different. We have clarified the same in the participants sub-section of Experiment 2.

- Results: Please report not only the significance but also the effect size for all the analyses.

- Response: Thank you for your comment. For the Friedman test, we calculated Kandall’s W as an index of the effect size. For the post-hoc analysis for the Friedman test and Wilcoxon signed-rank tests, the Z-score, previously reported in Tables 4 and 7, was divided by √n(n=sample size) as an index of the effect size. This value is a nonparametric version of Pearson’s r correlation. The values have been described in the results sections and the tables related to the analysis.

- Figure 3: There is a mistake. It should be T2 and T3 and not C2 and C3.

- Response: Thank you for bringing this to our attention. We have revised the letters in figure 3.

Reviewer #2

*Thank you for your valuable comments. Parts of the results in the previous manuscript were mixed with old version of manuscript. We apologize for the inattentiveness. We have revised the results section to ensure that there are no errors in the numerical values. No significant changes were made to the results and discussion sections.

Major comments:

1. Frankly, rather unequal sample sizes along with a just-significant difference in a key comparison in the experiment with the larger sample (exp.2, L2-T2) do not make me confident that these results are robust. I would feel much more at ease if there was some rationale for these sample sizes, as well as a more even discussion of the findings in experiment 2, i.e., not just of the significant difference L2-T2, but also of what the non-significant difference L3-T3 might mean.

- Response: Thank you for your comment. The size of sample size for the Friedman test in experiment 2 was calculated based on the parameters from experiment 1. The effect size of experiment 1 was 0.406 (Kendall’s W) and the nonsphericity correction epsilon was 0.687 (we used Greenhouse-Geisser epsilon for the conservative correction). The sample size was calculated by Gpower to achieve the power of 0.8 and the result was 26. Therefore, we set a goal to recruit 30 participants for experiment 2. However, due to the exclusion of participants who did not meet the analysis criteria, the final sample size was 29, which was still over 26. 

Regarding the robustness of the key comparison, we added the effect size of the Friedman test. The effect size of results in experiment 2 was 0.599, which falls in the moderate strength of effect size. Furthermore, the results of the post-hoc tests showed effect size (|Z|/√N, N=sample size) of 0.617 at T2-L2 comparison which is the nonparametric version of Pearson’s r correlation. Value over 0.5 is considered to be a large effect size. Based on these indices, we think that the results of experiment 2 can be considered quite robust. The discussion about the robustness of the anisotropy was added to the general discussion section (line 352-376). The line numbers are based on the clear version of the manuscript.

We also balanced the discussion about the results in the general discussion section. The discussion about the non-significant results of L3-T3 was added in line 332-334 of the clear version of the manuscript. We interpreted the results that 3cm of spatial incongruence was sufficient to abort BOI in both the transverse and longitudinal axis. 

2. Perhaps I missed it, but I would ask the authors to add a discussion of the difference between the two experiments. Notably, the much less clear anisotropy in the second experiment - meant to be a methodological improvement over the first - should be explained or at least touched upon. I think such a discussion on a more general level is missing and would add more value than the present discussion of many differences between single items and conditions (ll. 311-325) that may or may not mean much and where to me, it was not always clear which experiment was referred to. If the data do not paint a clear picture that is not itself a problem, but it needs to be made clear.

- Response: Thank you for your comment. We have added the procedure section for experiment 2 to describe the differences. In addition, we have explained the difference between experiment 1 and experiment 2 in the first paragraph of the general discussion section. The major difference was the hand posture of participants. In experiment 1, participants laid their hands flat on the table. However, this posture can cause an underestimation of distance perception in the longitudinal axis due to the shortcomings of the VR environment. For the accurate distance perception in the longitudinal axis, the depth cues should be presented like in a real-world scene. Due to the technological issue, participants in VR did not experience the full gamut of depth cues that were available in viewing a real world. The underestimation of depth in VR has been reported in previous studies (20-22). To exclude the confounding factor, participants raised their hands from the table and looked down at their own hands vertically. The horizontal line of sight was perpendicular to the longitudinal axis of the hands.

The discussion about the less clear anisotropy in the second experiment, despite the methodological improvement, was added in the general discussion section (line 352-376). The line numbers are based on the clear version of the manuscript. Since the possibility of distance underestimation in the longitudinal axis was eliminated in experiment 2, the score of L2 and L3 dropped and variance was increased. The variance increase could have originated from the individual difference in 2 point-discrimination threshold.

Although the anisotropy appeared less prominent due to the removal of the confounding factor regarding underestimation of depth, the anisotropy was reproduced in experiment 2 with moderate strength of effect size. We think this reproduction strengthens the point that the anisotropy did not result due to the VR depth underestimation. 

Lastly, the comparisons of scores between the single items were revised to be clearer for understanding. We described that the discussion of differences in single items of the questionnaire is referring to experiment 2 (line 377). The line numbers are based on the clear version of the manuscript.

Minor/specific comments:

3. The fact that bias and perception of incongruence were assessed with a questionnaire took me a bit by surprise after reading the abstract and introduction. Given that different tasks are discussed at length early on, it might help clarifying what each of those measures specifically.

- Response: Thank you for your comment. We have revised the questionnaire section by adding the explanation of the measurements for each item in Table 1. Explanation about what Q2 measures were revised to the measurement of referral of touch which appears more appropriate and the change was reflected in the discussion. 

4. How long was each experiment? Was there one trial per condition, or multiple?

- Response: Thank you for your comment. We described the length of each experiment and the trial per condition in the procedure section. Both experiments lasted approximately 45 minutes until the procedure was complete. There was one trial per condition.

5. Providing mean scores per item, with variability, would be much preferred over the pairwise comparisons for each question that are currently given in tables 4 and 7.

Similarly, these scores should be provided in the data file, which currently contains only aggregate values.

- Response: Thank you for your comment. We added the table about mean scores and variability per item in the manuscript. There were typing mistakes in the Q2 score in the longitudinal condition. We reanalyzed the Wilcoxon signed-rank test per item and confirmed that there was no change in the significance and non-significance of the results. The updated data was recently uploaded to Mendeley (link: https://data.mendeley.com/datasets/257zj3m2sr/1, DOI: 10.17632/257zj3m2sr.1).

6. P-values of precisely 0 or 1 should be reported as > .001 and < .999, respectively.

- Response: Thank you for your comment. We have edited the p-value of precisely 0 or 1 to > .001 and < .999 in the results section.

---

## [Decision Letter · Decision Letter 1]

27 Jun 2022

PONE-D-22-04614R1Persistence of Metric Biases in Body Representation during the Body Ownership IllusionPLOS ONE

Dear Dr. Lee,

Thank you for submitting your manuscript to PLOS ONE. After careful consideration, we feel that it has merit but does not fully meet PLOS ONE’s publication criteria as it currently stands. Therefore, we invite you to submit a revised version of the manuscript that addresses the points raised during the review process. Reviewer 2 has one final minor suggestion. I am giving you the chance to incorporate this small change to the abstract, but I don't anticipate the need for further review.  

We look forward to receiving your revised manuscript.

Kind regards,

Guido Maiello

Academic Editor

PLOS ONE

Journal Requirements:

Reviewers' comments:

Reviewer's Responses to Questions

**Comments to the Author**

1. If the authors have adequately addressed your comments raised in a previous round of review and you feel that this manuscript is now acceptable for publication, you may indicate that here to bypass the “Comments to the Author” section, enter your conflict of interest statement in the “Confidential to Editor” section, and submit your "Accept" recommendation.

Reviewer #2: (No Response)

2. Is the manuscript technically sound, and do the data support the conclusions?

Reviewer #2: Yes

3. Has the statistical analysis been performed appropriately and rigorously? 

Reviewer #2: Yes

4. Have the authors made all data underlying the findings in their manuscript fully available?

Reviewer #2: Yes

5. Is the manuscript presented in an intelligible fashion and written in standard English?

Reviewer #2: Yes

6. Review Comments to the Author

Reviewer #2: The manuscript has been thoroughly revised and much improved.

My one remaining issue is that it should be clear from the abstract that the main measure in this study was a questionnaire. Other than that, I have no more comments.

7. PLOS authors have the option to publish the peer review history of their article (what does this mean?). If published, this will include your full peer review and any attached files.

Reviewer #2: No

---

## [Author Response · Author response to Decision Letter 1]

8 Jul 2022

Response to Reviewers

Reviewer #2

- My one remaining issue is that it should be clear from the abstract that the main measure in this study was a questionnaire.

: Thank you for your comment. We added the lines to the abstract about the usage of a questionnaire as the main measurement method in the study (line 37-38).

---

## [Editor Report · Decision Letter 2]

13 Jul 2022

Persistence of Metric Biases in Body Representation during the Body Ownership Illusion

PONE-D-22-04614R2

Dear Dr. Lee,

We’re pleased to inform you that your manuscript has been judged scientifically suitable for publication and will be formally accepted for publication once it meets all outstanding technical requirements.

Kind regards,

Guido Maiello

Academic Editor

PLOS ONE
---

## [Editor Report · Acceptance letter]

15 Jul 2022

PONE-D-22-04614R2 

Persistence of Metric Biases in Body Representation during the Body Ownership Illusion 

Dear Dr. Lee:

I'm pleased to inform you that your manuscript has been deemed suitable for publication in PLOS ONE. Congratulations! Your manuscript is now with our production department. 

Kind regards, 

on behalf of

Dr. Guido Maiello 

Academic Editor

PLOS ONE